# Exploring Aβ Proteotoxicity and Therapeutic Candidates Using *Drosophila melanogaster*

**DOI:** 10.3390/ijms221910448

**Published:** 2021-09-28

**Authors:** Greta Elovsson, Liza Bergkvist, Ann-Christin Brorsson

**Affiliations:** 1Division of Molecular Biotechnology, Department of Physics, Chemistry and Biology, Linköping University, 58183 Linköping, Sweden; greta.elovsson@liu.se; 2Department of Neurobiology, Care Sciences and Society, Karolinska Institute, 17164 Solna, Sweden; lizabergkvist@gmail.com

**Keywords:** Alzheimer’s disease, amyloid-β peptide, proteotoxicity, drug candidates, *Drosophila melanogaster*

## Abstract

Alzheimer’s disease is a widespread and devastating neurological disorder associated with proteotoxic events caused by the misfolding and aggregation of the amyloid-β peptide. To find therapeutic strategies to combat this disease, *Drosophila melanogaster* has proved to be an excellent model organism that is able to uncover anti-proteotoxic candidates due to its outstanding genetic toolbox and resemblance to human disease genes. In this review, we highlight the use of *Drosophila melanogaster* to both study the proteotoxicity of the amyloid-β peptide and to screen for drug candidates. Expanding the knowledge of how the etiology of Alzheimer’s disease is related to proteotoxicity and how drugs can be used to block disease progression will hopefully shed further light on the field in the search for disease-modifying treatments.

## 1. Introduction

Neurodegenerative diseases, such as Alzheimer’s disease (AD), are associated with proteotoxicity, which is caused by protein aggregation and results in extensive neuronal damage in the brain. Neurodegeneration and the disturbance of essential functions in the cell manifest in cognitive impairments and premature death [1,2]. A main proteotoxic contributor in AD is the amyloid-β (Aβ) peptide, where the propensity to aggregate differs between different variants [3]. Aβ misfolds and aggregates into pre-fibrillar assemblies that are highly associated with toxicity. They then progressively merge into mature fibrils [1,4]. There is an urgent need to find disease-modifying treatments and, therefore, reliable and powerful methods to pin down fundamental components underlying the mechanisms responsible for proteotoxicity are needed. *Drosophila melanogaster* has proved to be a dynamic model organism that can produce high-quality data in a short time frame. One of the fly’s most prominent feature is the possibility to perform genetic alterations through the well-known Gal4/UAS expression system, thus making it possible to express target proteins in a specific cell type or tissue [5]. In addition, *Drosophila* offers a number of advantages as a model system for studying diseases where: (i) a variety of phenotypic markers are available for identifying detrimental effects due to proteotoxicity, (ii) the lifespan of *Drosophila* makes it possible to investigate age-related diseases on a reasonable time scale (days to weeks, as opposed to months and years, in mouse model systems), (iii) the system is amenable to large drug screens since the flies proliferate well and are relatively inexpensive and easy to work with and (vi) there are extensive tools that allow disease-related genes and molecular pathways to be genetically and pharmacologically manipulated in order to find out both the function of their orthologs in vivo, and how these genes are involved in the pathogenesis of different diseases, which can generate in vivo data that are translatable to mammalian system [6,7]. In this review, we outline phenotypes and assays that may be assessed and applied to examine proteotoxicity in *Drosophila* and present different *Drosophila* models to study proteotoxicity caused by the aggregation of Aβ. We also show how AD fly models have been used to find drugs with anti-proteotoxic effects and we elucidate the protective mechanisms of these drugs in *Drosophila*.

## 2. Direction of Protein Expression to *Drosophila* Neurons

The first transgenic *Drosophila* strain was created in the early 1980′s when Rubin and Spralding described the use of the P-element (i.e., transposon) technique [8]. Brandon and Perrimon further developed this method in 1993, resulting in the now widely used Gal4/UAS expression system [5]. This system requires two separate fly lines; the first is called a reporter line, carries the genes of interest and is placed downstream of an upstream activating sequence (UAS). For the gene to be transcribed, a transcriptional activator, in this case Gal4, must bind to the UAS domain. In the absence of Gal4, the gene is silent. The second fly line needed is called the driver line, which continuously expresses the transcriptional activator Gal4. By placing the gene encoding Gal4 downstream of a cell- or tissue-specific promoter, the expression of the Gal4 protein will be directed to a certain cell or tissue type. When the reporter line is crossed with the driver line, the generated offspring will be transgenic for the genes of interest, as well as expressing Gal4, resulting in the cell- or tissue-specific transcription and translation of the protein of interest. 

One commonly used driver line that directs protein expression to neuronal cells is the embryonic lethal abnormal vision (ELAV)-Gal4 [9]. Previously believed to direct the protein expression of the gene of interest in neuronal cells exclusively, later research has shown ELAV to have a broader expression pattern than expected, driving protein expression in neural progenitor cells and nearly all embryonic glial cells [10]. Similarly to ELAV-Gal4, the neuronal synaptobrevin (nSyb)-Gal4 driver is considered to be pan-neuronal. However, while ELAV-Gal4 generates the highest expression levels in early developmental stages, with less expression in adults, nSyb-Gal4 expression is stronger in adult flies. Thus, the choice of driver line should be based on experimental needs and research question. 

The glass multimer reporter (GMR)-Gal4 is commonly used to direct protein expression to the developing eye in all cell types, posterior to the morphogenetic furrow [11]. This allows for a quick and easy evaluation of the potential proteotoxicity in the form of a rough eye phenotype, discussed in detail below. However, as has been shown for other driver lines, the GMR expression pattern is broader than originally believed, with additional expression observed in the brain, trachea and leg discs [12].

## 3. Methods to Study Proteotoxicity in *Drosophila*

A variety of methods have been developed to investigate the proteotoxicity in *Drosophila*. Proteotoxicity affects cellular functions and triggers certain events, which may give rise to phenotypes that can be assessed to quantify toxicity in the fly. A graphical overview of phenotypes that can be studied to examine proteotoxicity in *Drosophila* is shown in Figure 1 and includes: the viability, cell death, cellular impact, protein levels, protein aggregates and cognition. These phenotypes and suggested methods to monitor them are described below. 

Viability (Figure 1A): A reliable method to assess the viability of the fly is a survival assay, which provides unambiguous facts about the fly’s health in general by analyzing the survivability variance between different populations of *Drosophila* [13,14,15]. The survival/death of the subjects in a fly cohort is registered continuously until all flies have died. Commonly, the value for the median survival, which is the time when 50% of all flies have died, is extracted from the survival curve and compared with other genotypes/groups. Another method used to analyze the viability is a locomotor assay, where the activity of the flies is monitored over time. In a contained space (e.g., a plastic vial), healthy flies instinctively move upwards in a straight line and, as the flies get older, their activity decreases. A common method to assess locomotor dysfunction is a climbing assay. Here, the flies are allowed to move upwards in a vial and, after a certain time, the number of flies at the top and bottom of the vial is counted. As the flies age or become sick, their ability to reach the top decreases. Two defined parameters that can be used to probe activity are the speed and angle of movement [14,15,16,17]. The average velocity for a healthy young fly is 10 mm/s and, as the fly ages or becomes sick, the velocity decreases until the fly becomes immobile. The angle of movement describes the degree of deviation from a vertical line when flies move from the bottom to the top of the vial. This value is approximately 55° for a healthy young fly and increases as the fly ages or becomes sick. 

Cell death (Figure 1B): A phenotype that can been used to probe cell death is the so-called rough eye phenotype [14,18,19]. This is a developmental phenotype that can occur if a toxic protein is expressed in the neurons of the fly eye using the GMR-Gal4 driver. A toxic protein will give rise to a distorted eye structure where the ommatidia are fused. This eye phenotype gives a straightforward image of the toxicity, since a higher degree of toxicity correlates with a more severe rough eye phenotype. Another method to probe cell death is the quantification of apoptotic cells using a terminal deoxynucleotidyl transferase dUTP nick end labeling-(TUNEL) assay [20,21]. In this assay, the enzyme terminal deoxynucleotidyl transferase (TdT) labels DNA breaks that occur during apoptosis, thus making it possible to identify apoptotic DNA fragmentation in the tissue using a histochemical readout. 

Cellular impact (Figure 1C): The presence of a toxic protein can disrupt critical pathways in the cell, which can result in various adverse conditions that can be studied to probe proteotoxicity. One example of an adverse condition is abnormal autophagy. Autophagy is a “self-eating” process in which cellular components undergo lysosomal breakdown and the acquired material is then recycled back into the system. Autophagy is an intuitive response to cellular stress due to e.g., nutrient scarcity and is associated with neurodegeneration and cancer if incorrectly regulated. *Drosophila* is an organism where autophagy plays an important role during several metamorphic phases, especially the starvation period, wherein the pupae is being developed into an adult fly. Changes in autophagy can be probed by using electron microscopy (EM) to distinguish autophagic vesicles, followed by confocal microscopy and biochemical assays for further examination [22,23]. Oxidative stress is another adverse condition that can be caused by the presence of a toxic protein, and results in neurodegeneration. Oxidative stress reflects an imbalance in the cell that results in increasing manifestations of reactive oxygen species (ROS) that can lead to cellular damage [24]. An indicator for oxidative stress is protein carbonylation, which is an irreversible protein modification that can be probed by immunoblot analysis [20]. 

Protein levels (Figure 1D): When studying proteotoxicity, it is of interest to investigate the levels of toxic proteins in the flies. This is commonly carried out by a Western blot, which is a useful method to analyze the presence of various proteins in a crude tissue sample by relying on specific antibodies for detection [14,25]. Additionally, an estimate of the protein concentration is generated by comparing the intensity of the migrated bands. If the protein concentration needs to be measured more accurately, a suitable method is the Meso Scale Discovery (MSD) platform [14,26]. MSD is an antibody-based technique that uses electrochemiluminescence for the detection of the protein. When preparing a fly sample for protein detection, it is possible to gather certain body parts of the fly depending on in which tissue the target gene has been expressed. In addition, the fractions of soluble and insoluble protein species can be extracted separately to analyze how the levels of soluble and insoluble protein species in the fly tissue are related to proteotoxicity. 

Protein aggregates (Figure 1E)*:* Protein misfolding diseases are associated with the accumulation of protein aggregates. However, the connection between the formation of a certain protein aggregate and toxicity is still not fully understood. The characterization of the presence of different protein aggregates and proteotoxicity can therefore provide valuable information about the features of the toxic protein species. Various compounds can be used to stain protein aggregates in vivo, including luminescent conjugated oligothiophenes (LCOs), such as p-FTAA [14,27,28] and h-FTAA [29], Congo Red [15,30], thioflavin-S [18,30,31] and different antibodies specific for a certain protein or aggregate morphology [15,20,31]. To visualize the cell nuclei, counterstaining can be performed using 4′,6-diamidino-2-phenylindole (DAPI) [27], a blue fluorescent DNA stain, or ToPro3 stain [28] with a far-red fluorescence spectrum.

Cognition (Figure 1F): For the untrained eye, *Drosophila melanogaster* seems to be a simple, unsophisticated organism, but the flies exhibit some advanced behaviors, such as courtship, learning and memory. Pavlovian conditioning is a test in which the flies are exposed to two odors, one of which is selectively avoided or selected due to previous experience of that odor with a negative (e.g., electric shocks) or positive (e.g., food with sugar) event [32,33]. In an additional test, courtship conditioning, the male courtship is suppressed as a result of the fly being continuously rejected by newly mated females [32]. For a short time after mating, females have no interest in mating again and are therefore unreceptive towards interested males. As a consequence, these males then suppress their courtship behavior (e.g., tapping) due to continuous rejection. This courtship suppression remains even after they are moved to virgin females that are interested in mating. Changes in these behaviors can be early signs of proteotoxic events in the flies.

## 4. *Drosophila* and Proteotoxicity of the Aβ Peptide

There are two crucial events that occur in AD, namely the formation of amyloid plaques and neurofibrillary tangles composed of the Aβ peptide and hyperphosphorylated tau, respectively [1,34,35]. Proteotoxic events, exerted by the Aβ peptide, are believed to play a major role in the pathological process of AD [3,36,37]. Aβ is yielded from AβPP through sequential cleavages by BACE1 and thereafter γ-secretase [38]. The length of Aβ depends on the location of the γ-secretase cleavage site and, consequently, this results in a variety of different isoforms, where Aβ1-40 and Aβ1-42 are most frequently produced. Aβ1-42 presumably plays the most important role in the disease progress due to its toxicity and high propensity to aggregate [39,40]. Therefore, it is not surprising that Aβ1-42 has been studied extensively for its proteotoxic effect in *Drosophila*.

### 4.1. The History of Drosophila Aβ Models

In 2004, groundbreaking work exploring Aβ toxicity in *Drosophila* was published [33,41]. There, the expression of Aβ1-42 in fly CNS using the Gal4/UAS system caused the formation of Aβ deposits accompanied by neurodegeneration, locomotor deficits and a shortened lifespan. Additionally, the photoreceptor-directed overexpression of Aβ1-42 resulted in rough eye phenotypes where the degree of eye disruption depended on the expressed amount of Aβ1-42 [41]. The following year, a similar model of Aβ1-42 proteotoxicity was published where the peptide was again expressed in the neurons using the Gal4/UAS system, but, in this model, a signal peptide was connected to Aβ1-42 to allow for secretion [15]. The proteotoxic effects of Aβ1-42 were implicated in a reduced viability, cell death and the accumulation of Aβ deposits in the flies. It was observed that locomotor deficits occurred before the formation of amyloid plaques and neurodegeneration, suggesting that intracellular, toxic oligomers of Aβ might be responsible for proteotoxicity. Moreover, the Arctic mutant (Glu22Gly) of the Aβ1-42 peptide was introduced in this study, which greatly reduced the viability, resulting in a significantly shorter lifespan for the Arctic Aβ1-42 flies compared to flies expressing wild type Aβ1-42 [15]. The Arctic mutation is found in rare cases of familial AD and is associated with an accelerated fibrillation rate of Aβ [42]. A recent study from 2020 using Arctic Aβ1-42-expressing flies identified that the levels of specific proteins were altered due to Aβ accumulation [43]. A common denominator for these proteins is that they are associated with the brain protein interaction network and assist in key molecular processes for cellular function, signaling and homeostasis. Thus, Aβ-induced alterations in the brain proteome may disrupt these processes and thereby cause AD pathology.

### 4.2. Investigating Aβ Isoforms in Drosophila

The shorter Aβ1-40 peptide has also been extensively investigated and, in contrast to Aβ1-42, has displayed relatively negligible toxic effects in *Drosophila* [15,33]. Despite the discovery of the age-dependent accumulation of SDS-soluble Aβ1-40 species in the fly brain, only learning defects were observed, and no signs of amyloid deposits or neurodegeneration were detected [33]. Since different isoforms of the Aβ peptide can be found in the AD brain, it is important to consider their possible triggering effect on each other, causing a seeding cascade followed by proteotoxicity. A study from 2017 examined whether insoluble and otherwise nontoxic Aβ1-40 could contribute to AD pathology due to seeded propagation by Aβ1-42. Through this study, it was revealed that even a small amount of Aβ1-42 seeds was sufficient to promote the formation of Aβ1-40 deposits and induce toxicity [44].

The probability of a protein aggregating depends on its local effective concentration. By linking two copies of the Aβ peptide in a tandem (head-to-tail) construct, the aggregation process was accelerated, resulting in an increased amyloid deposition and proteotoxicity in tandem Aβ1-42 expressing flies compared to flies expressing two copies of single Aβ1-42 [45]. Expressing a tandem construct of Aβ1-40 increased the aggregation process but did not cause any toxic effect in *Drosophila*. Data from the study revealed that the propensity to populate soluble oligomeric species is higher for tandem linked Aβ1-42 compared to tandem linked Aβ1-40, and it is likely that these soluble oligomeric species are responsible for the proteotoxic effect detected in the flies. The ability of Aβ1-40 to aggregate into mature fibrils without an extensive population of soluble oligomeric species might explain the lack of toxicity for the tandem Aβ1-40 flies.

One study investigated the proteotoxic behavior of different Aβ isoforms in *Drosophila*, including Aβ1-36 to Aβ1-40, Aβ1-42 and Aβ1-43 [19]. Among these peptides, the Aβ1-42 showed the highest toxicity, which manifested in impaired locomotor activity along with a strong rough eye phenotype. A proteotoxic effect was found for the Aβ1-43 peptide, albeit to a lower extent compared to the Aβ1-42 peptide. The Aβ1-36 to Aβ1-40 did not show any significant toxic effect in the flies and, when co-expressed with Aβ1-42, they were found to partially attenuate Aβ1-42 toxicity in a dose-dependent manner, indicating that these shorter peptides may counteract the pathological progress of AD [19]. Another study found that Aβ1-43 triggered a toxic effect of Aβ1-40 in flies, resulting in impaired climbing abilities and premature death [46]. The lower proteotoxic effect of Aβ1-43 compared to Aβ1-42 has been confirmed in other *Drosophila* studies [28,46] and the nontoxic observation of C-terminally truncated Aβ-peptides (Aβ1-37 to Aβ1-41) has previously been observed [28]. In that study, N-terminally truncated Aβ variants were also examined, where Aβ3-42 exhibited a similar toxicity to Aβ1-42, whereas the toxicity of Aβ11-42 was found to be lower compared to Aβ1-42. By exploring the proteotoxicity of various mutated Aβ isoforms, the study found that the N-terminal mutation E11A of the Aβ11-42 peptide and the C-terminal mutations A42D, A42R and A42W of the Aβ1-42 peptide had a reduced effect on the proteotoxicity. In conclusion, these results reveal the importance of E11, A42 and the first 10 amino acids in the Aβ sequence in achieving a full proteotoxic effect of the Aβ1-42 peptide.

Besides the “regular” variants of Aβ, studies on modified Aβ variants have also been made. N-terminally modified Aβ exists in the brain of an AD patient, where the most prevalent form is the pyroglutamate-modified Aβ (AβpE3-42) peptide. A publication from 2016 has described the effects of AβpE3-42 in *Drosophila* and observed that AβpE3-42 was more toxic than Aβ1-42 and that AβpE3-42 enhanced the toxicity of Aβ1-42. When Aβ1-42 was co-expressed with AβpE3-42, the levels of Aβ1-42 increased significantly, leading to an increased proteotoxicity [47]. This suggests that AβpE3-42 can seed the aggregation of Aβ1-42 and thereby induce Aβ-mediated proteotoxicity. A summary of Aβ isoforms and their relative toxicity can be found in Figure 2.

### 4.3. The AβPP-BACE Fly

Another approach to study Aβ proteotoxicity in *Drosophila* is to generate the production of the Aβ peptide through the processing of AβPP by BACE1 and by endogenous fly γ-secretase [14,20,31,48,49]. Studies using this AβPP-BACE1 *Drosophila* model of AD have shown that the toxic effect per amount of detected Aβ1-42 in the fly is higher when the peptide is produced by AβPP processing compared to when expressed directly from the transgene. However, the toxicity found in the AβPP-BACE1 flies cannot be attributed solely to the Aβ1-42 peptide, since a wide range of different cleavage products are formed from AβPP processing, as well as post-translationally modified Aβ isoforms. Thus, the AβPP-BACE1 *Drosophila* model of AD cannot be used for the purpose of studying proteotoxic effects caused by a specific Aβ variant, but rather to examine how AβPP processing affects the flies.

### 4.4. Relative Toxicity between Aβ Isoforms

There appears to be a connection between the degree of proteotoxicity in *Drosophila* models of different Aβ peptides and their aggregation behavior. Generally, in all studies, the Arctic mutant (Glu22Gly) of Aβ1-42 is more toxic than Aβ1-42, which correlates to the higher ability of Arctic Aβ1-42 to form pre-fibrillar species compared to Aβ1-42. Additionally, the 42nd amino acid (A42) in the Aβ1-42 sequence seems to be important for the propensity of the peptide to aggregate, since mutations at this location reduce Aβ accumulation and toxicity. This could explain why Aβ1-40 is not as aggregation-prone as Aβ1-42 and does not contribute to the AD pathology to the same extent as the Aβ1-42 peptide. It is difficult to rank the degree of proteotoxicity of Aβ variants, which have been studied in different publications, but a rough estimate of the relationship between the toxicity of different Aβ variants, examined in *Drosophila*, is illustrated in Figure 2. Unravelling the importance of different Aβ variants, along with how specific amino acids contribute to proteotoxicity, is valuable information needed to enhance our knowledge about the disease mechanism behind AD.

## 5. *Drosophila* as Model Organism for Drug Screen against Aβ Proteotoxicity

There are two main approaches to delivering the drug when using *Drosophila* to screen for compounds against proteotoxicity. The drug molecule can be mixed in the food and administrated to the fly expressing the proteotoxic protein, or, if the drug is a protein, it can be co-expressed with the proteotoxic protein in the fly.

### 5.1. Blocking Aβ Aggregation

In one of the first *Drosophila* models of AD, the amyloid-binding dye Congo Red was tested for protection against the proteotoxicity of the Aβ peptide [15]. Feeding the flies food mixed with Congo Red extended the life span of both Aβ1-42 and Arctic (Glu22Gly) Aβ1-42-expressing flies. Histology experiments revealed fewer protein aggregates in Aβ1-42-expressing flies treated with Congo Red compared to untreated flies. This study shows that the proteotoxic effect of the Aβ peptide can be hindered in vivo by a drug that blocks the aggregation process of the peptide. Indeed, considering the connection between the formation of toxic Aβ species and AD, finding a drug that inhibits the Aβ aggregation process should be an effective therapeutic strategy. Using this approach, a synthetic molecule designated D737 (C25H20N2O) was discovered when a library of compounds was screened for anti-Aβ aggregation properties [50]. The compound D737 increased the life span and improved the climbing performance of both Aβ1-42 and Arctic Aβ1-42-expressing flies [51]. In a follow up study, two analogs of D737 with anti-Aβ aggregation properties (D744 and D830) were identified. These analogues were able to rescue Aβ proteotoxicity more efficiently than D737, thus strengthening the evidence of a correlation between the anti-Aβ aggregation properties of a drug and its ability to block Aβ toxicity in vivo. Another study found that feeding AD flies with acetylcholinesterase inhibitors improved the longevity and mobility of Arctic (Glu22Gly) Aβ1-42 flies and that the number of aggregates in the fly brain was reduced [52]. This rescue effect was particularly evident for the newly synthesized acetylcholinesterase inhibitor XJP-1 and was attributed to the ability of XJP-1 to inhibit the acetylcholinesterase-induced aggregation of the Aβ peptide.

### 5.2. Enhancing Aβ Aggregation

In contrast, when the effect of curcumin on Aβ proteotoxicity was investigated in *Drosophila*, it was found that the toxicity of the Aβ peptide can be suppressed by enhancing the fibrillation process [27]. In this study, flies overexpressing Aβ1-42 or the Arctic (Glu22Gly) Aβ1-42 variant were fed with a substrate mixture containing curcumin. Although it was found that curcumin by itself is somewhat toxic to the flies, survival and locomotor analyses showed a rescue effect for the Aβ expressing flies treated with curcumin. Histochemistry analyses revealed that the presence of curcumin accelerated the Aβ fibrillation process in the fly brain, thereby reducing the pool of toxic prefibrillar species. In non-transgenic *Drosophila*, curcumin has shown to both down-regulate the gene expression of acetylcholinesterase, thereby increasing neuronal signaling, and to improve the antioxidant status [53]. These factors might contribute to the anti-toxic effect of curcumin in AD flies.

### 5.3. Increasing Protein Clearance

A compound that can increase the ability of the cells to degrade the Aβ peptide would be an interesting drug candidate. Indeed, this year, the FDA approved aducanumab, the first monoclonal antibody treatment for AD (Biogen). This approach has also been investigated in *Drosophila*, where an engineered Aβ binding affibody protein was co-expressed with the Aβ peptide in the fly brain [54]. The affibody molecule consists of a three-helix Z domain and can be selected for different binding properties using phage display. The presence of two copies of the affibody protein, connected head-to-tail, resulted in an impressive increase in the lifespan for both Aβ1-42 and Arctic (Glu22Gly) Aβ1-42-expressing flies. In addition, the abnormal rough eye phenotype of the Arctic Aβ1-42-expressing flies was suppressed. Biochemical analyses showed that the Aβ levels and deposits of Aβ aggregates in the fly brain decreased sharply, indicating that the anti-Aβ proteotoxic effect of the affibody protein is due to its ability to promote the clearance of the peptide in fly tissue. In a recent study, Aβ1-42-expressing flies were fed with extract from red adzuki beans [55]. Data from this study showed a rescue effect for the Aβ flies, which manifested in an increased longevity and locomotor activity and an in memory improvement of the adzuki-bean-treated flies. In addition, the Aβ level of the treated Aβ flies decreased compared to the untreated flies, indicating that the intake of red adzuki beans improves the degradation process of the Aβ peptide in the fly brain, which protects the neurons against Aβ proteotoxicity. A similar result was found for the protein puromycin-sensitive aminopeptidase (SPA) that was tested for anti-toxic effects in AD flies [56]. The co-expression of this enzyme in Arctic Aβ1-42 flies resulted in an increased life span and activity and the Aβ-induced rough eye phenotype was rescued. Additionally, the Aβ levels and deposits of Aβ in the fly brain were greatly reduced in the presence of SPA, suggesting that the rescue effect is due to the enzyme’s ability to enhance clearance of the peptide.

### 5.4. Proteins and Peptides as Drug Candidates

The advantage of testing a protein as an anti-proteotoxic drug in fly models is that the drug–protein can be co-expressed with the proteotoxic protein, ensuring that the two molecules will be present in the fly tissue simultaneously. This way, the lysozyme protein was tested for its anti-Aβ proteotoxic effect in AD flies [57,58]. Co-expressing lysozyme with Aβ1-42 extended the life span and improved the activity of the flies. In addition, the rough eye phenotype in Aβ1-42-expressing flies was suppressed. Lysozyme was found to interact with the Aβ1-42 peptide in vivo and to reduce the Aβ levels in the fly brain. These data suggest that the anti-toxic effect of lysozyme is due to its ability to disrupt the Aβ aggregation process, resulting in non-toxic species, and facilitating the Aβ degradation process.

Interestingly, shorter Aβ peptides can counteract Aβ proteotoxicity [19]. The co-expression of Aβ1-36 to Aβ1-39 peptides with Aβ1-42 did partially rescue the locomotor dysfunction and rough eye phenotype of Aβ1-42 flies. The distribution of Aβ assemblies in the mushroom body neurons of the flies was not affected by the presence of the shorter peptides and no apparent correlation was found between their rescue effects and the Aβ1-42 levels in the flies. A proposed protection mechanism for these shorter peptides is that they interfere with the Aβ1-42 aggregation process in various ways, with the common result that the level of toxic Aβ species in the fly is reduced.

Overexpressing the chaperon domain proSP-C BRICHOS was found to protect AD flies from Aβ proteotoxicity [59,60]. The rescue effects were manifested in the extended lifespan and increased locomotor activity when proSP-C BRICHOS was co-expressed with Aβ1-42 in the fly neurons. In addition, the deposition of Aβ aggregates in the fly brain was delayed and the ratio between soluble and insoluble Aβ was increased. Later, a study was published where the anti-Aβ proteotoxic effects of the proSP-C BRICHOS and Bri2 BRICHOS domains were investigated in parallel. The study showed that the Bri2 BRICHOS domain can prevent Aβ1-42 toxicity in the flies in a similar fashion as the proSP-C BRICHOS domain, albeit more efficiently. Additionally, a rescue effect of the eye phenotype was also confirmed. In vitro analyses revealed that the BRICHOS domains inhibit the aggregation process of Aβ but in different ways. Whereas proSP-C BRICHOS specifically affects the secondary nucleation event [61], Bri2 BRICHOS inhibits the aggregation in a more comprehensive way, affecting both the secondary nucleation and fibril-end elongation. Both BRICHOS domains interfere with the Aβ aggregation in such a way that the formation of toxic Aβ species is reduced, which slows down the disease progression in the flies.

### 5.5. Targeting Inflammatory Processes

It is well known that inflammation is a prominent feature in AD [62]. Thus, one therapeutic strategy is to find natural or synthetic drugs with anti-inflammatory properties. With this in mind, Aβ1-42-expressing flies were treated with an extract from the plant *Arabidopsis thaliana* known to contain polyphenols, which is a group of natural compounds that possess anti-inflammatory properties [63]. The extract was found to increase the activity of the flies when assessed by a climbing assay. Among the polyphenol compounds that were detected in the extract from *Arabidopsis thaliana*, two derivates of kaempferol and quercetin were identified, as well as luteolin. These substances have been tested separately for their anti-Aβ proteotoxic effect in AD flies. Quercetin was found to extend the lifespan and increase the activity of Arctic (Glu22Gly) Aβ1-42-expressing flies [64]. In this study, a detailed transcriptomic analysis revealed the disturbance of cell signaling pathways in the AD flies. Specifically, the expression of proteins involved in the FoxO cell cycle signaling pathway and in DNA replication was found to be dysregulated in the AD flies, which most likely contributes to toxicity. These pathways were largely restored by the presence of quercetin in the fly brain, which indicates that the anti-Aβ proteotoxicity mechanism can be attributed to the compound’s ability to re-establish cell signaling pathways and DNA replication. Feeding Aβ1-42-expressing flies with kaempferol increased the climbing performance and protected the AD flies from memory loss and oxidative stress. In addition, the compound reduced the acetylcholinesterase activity, which increases neuronal signaling [65]. Luteolin was found to rescue Aβ1-42 toxicity in longevity and climbing assays, and the formation of Aβ aggregates in the AD fly brain was reduced [66]. Moreover, the acetylcholinesterase activity and oxidative stress were suppressed in the luteolin-treated AD flies. In summary, in addition to anti-inflammatory characteristics, the protection of *Arabidopsis thaliana* extract against Aβ toxicity is likely due to a combination of anti-Aβ proteotoxic effects exerted by various polyphenols, which includes a decrease in the formation of Aβ aggregates, a decrease in acetylcholinesterase activity, the restoration of cell signaling pathways for the cell cycle and DNA replication and protection against oxidative stress.

Treating Aβ1-42-expressing flies with extracts from *Gardenia jasminoides* did also rescue Aβ toxicity without any detectable interfering with the Aβ aggregation process [67]. The anti-proteotoxic effect of the extract was manifested by preventing memory loss of the AD flies. Quantifying the level of soluble and insoluble Aβ levels in the flies did not show any differences between treated and non-treated Aβ flies, leading to the conclusion that the rescue effect was not due to any changes in the Aβ aggregation process. Instead, the study showed that the rescue effect of the extract was due to its capacity to downregulate the expression of inflammatory genes that were found to be upregulated, causing toxicity in the Aβ flies.

### 5.6. Preventing Oxidative Stress

Reducing oxidative stress could be a very important target for a therapeutic strategy against AD, since the disease is thought to be accompanied by an excessive production of ROS, leading to cell death [24]. Interestingly, feeding Aβ1-42-expressing flies with nordihydroguaiaretic acid (NDGA), which possesses both antioxidant and free radical scavenging properties, extended the life span and increased the climbing ability of the Aβ flies [68]. A delay in the memory loss was also detected for the NDGA-treated Aβ flies, and oxidative stress and the acetylcholinesterase activity were reduced. The deposition of Aβ in the fly brain was not affected by the presence of NDGA, which indicates that the anti-Aβ proteotoxic effect of NDGA does not involve the disruption of the Aβ aggregation process but is rather exerted by the ability of NDGA to both increase neuronal signaling and reduce the formation of ROS in the AD flies.

### 5.7. Preventing Mitochondrial Dysfunction

Mitochondrial dysfunction is associated with AD [69] and could thus be a relevant target when developing therapeutic strategies [37]. This area has been explored in a study where AD flies that overexpressed a tandem variant of the Aβ1-42 peptide were treated with a compound named GMP-1 that is able to counteract mitochondrial dysfunction [70]. Expressing a tandem repeat of the Aβ1-42 peptide increases the ability of the peptide to form oligomeric aggregates and boosts the toxic effect in the flies [45]. Using this AD model, the GMP-1 compound was tested for anti-Aβ proteotoxic effects in vivo. A neuroprotective effect was detected where the longevity and climbing behavior were improved for the GMP-1 treated flies, which was attributed to the ability of GMP-1 to restore the mitochondrial function in the AD flies.

Figure 3 shows an overview of the different protective mechanisms of drugs with an anti-Aβ proteotoxic effect in *Drosophila*. These drug tests in *Drosophila* reveal that there might be several approaches to finding a treatment for AD. Most likely, a mixture of drugs with different modes of action would be necessary to block the disease progress and to ultimately cure the disease.

## 6. Conclusions

Over the past two decades, *Drosophila* has been extensively used to study the disease mechanism behind protein aggregation and neurodegeneration for several protein misfolding diseases. In this review, we have focused on the use of AD fly models to investigate the proteotoxic effects of different Aβ isoforms, as well as to search for compounds that can counteract this toxicity. Although there are differences in the proteome between a fly brain and a human brain that might limit the possibility of directly applying results from fly experiments to humans, the fly has proven to be a powerful tool to unravel Aβ-related proteotoxicity and to find potential drug candidates. Thus, data from various fly studies of Aβ proteotoxicity are likely to make a significant contribution to ultimately finding a therapeutic strategy to cure AD.

## Figures and Tables

**Figure 1 ijms-22-10448-f001:**
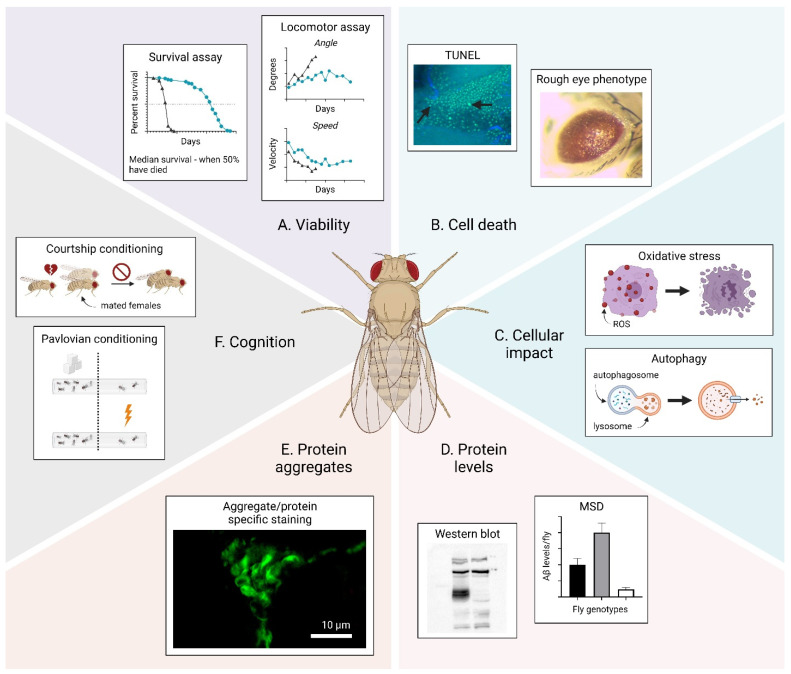
Overview of phenotypes that can be assessed to examine proteotoxicity in *Drosophila*. (**A**) Viability. (**B**) Cell death. (**C**) Cellular impact. (**D**) Protein levels. (**E**) Protein aggregates. (**F**) Cognition. The arrows in (**B**) show apoptotic cells. The figure was generated using BioRender.

**Figure 2 ijms-22-10448-f002:**
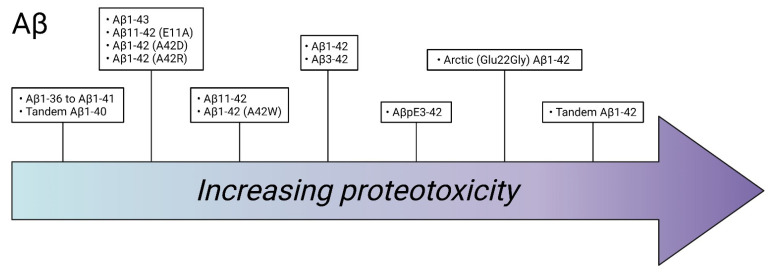
Estimated proteotoxicity for amyloid-β (Aβ) variants examined in *Drosophila*. Variants with a similar degree of toxicity are grouped together. The figure was generated using BioRender.

**Figure 3 ijms-22-10448-f003:**
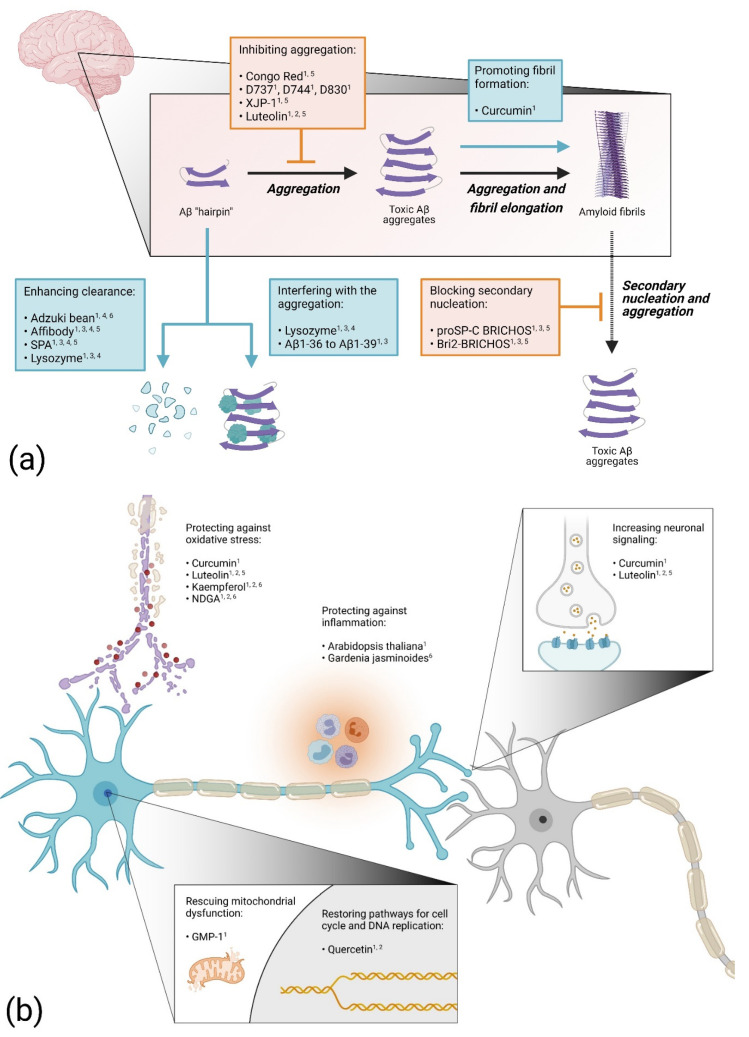
Overview of suggested anti-Aβ proteotoxicity mechanisms for different drugs examined in *Drosophila*: (**a**) drugs acting on the aggregation process; (**b**) drugs acting on cellular functions. Elevated numbers indicate confirmed protected effect in Alzheimer’s disease (AD) flies: ^1^ Improved viability; ^2^ Prevented adverse cellular impact; ^3^ Blocked cell death; ^4^ Reduced protein levels; ^5^ Reduced protein deposits; ^6^ Improved cognition. The figure was generated using BioRender.

## Data Availability

Not applicable.

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
