# Peer review of "Exploring Aβ Proteotoxicity and Therapeutic Candidates Using Drosophila melanogaster"

_ijms, 2021, doi:10.3390/ijms221910448_

Round 1

Reviewer 1 Report

General comments

This is an excellent review of utmost value to the field. Your conclusive work can fascilitate the progress of research and broadly explores proteiotoxicity of Aβ and drug testing using Drosophila melanogaster as a model organism for fasts detecting and evaluation of efficacy.

However, I ask you kindly to add more information on drug and nutraceutical development in the field as important information there on has become available recently (see for instance but not only):

Uras et al., 2021, In vivo evaluation of a newly synthesized acetylcholinesterase inhibitor in a transgenic Drosophila model of Alzheimer´s disease.

Jalali et al., 2021, Nutraceuticals and probiotic approaches to examine molecular interactions of the amyloid precursor protein APP in Drosophila models of Alzheimer´s disease.

Recently, also many new findings have been published on the molecular mechanisms and mediators of neuroprotection (see for instance but not only):

Tsakiri et al., 2021 Amyloid toxicity in a Drosophila Alzheimer´s model is ameliorated by autophagy activation.

Also classical work like that of

Chakraborty et al., 2011, Characterization of a Drosophila alzheimer´s disease model: pharmacological rescue of cognitive defects.

needs to be discussed.

Thank you for refering to these papers and the many important references cited therein. Please discuss the hot topics therein briefly in addition to the other aspects already covered.

Specific suggestions

Line 279: Error! Reference source not found. This bold insert can be deleted.

Line 442: Same as above

Author Response

Reviewer 1:

This is an excellent review of utmost value to the field. Your conclusive work can facilitate the progress of research and broadly explores proteotoxicity of Aβ and drug testing using Drosophila melanogaster as a model organism for fasts detecting and evaluation of efficacy. However, I ask you kindly to add more information on drug and nutraceutical development in the field as important information there on has become available recently (see for instance but not only):

Uras et al., 2021, In vivo evaluation of a newly synthesized acetylcholinesterase inhibitor in a transgenic Drosophila model of Alzheimer´s disease.

Author response: This reference has been added in the manuscript. Line 307:Another study found that feeding AD flies with acetylcholinesterase inhibitors improved the longevity and mobility of Arctic (Glu22Gly) Aβ1-42 flies and the number of aggregates in the fly brain was reduced (Uras et al., 2021). This rescue effect was particularly evident for the newly synthesized acetylcholinesterase inhibitor XJP-1 and was attributed the ability of XJP-1 to inhibit acetylcholine induced aggregation of the Aβ peptide.”

Jalali et al., 2021, Nutraceuticals and probiotic approaches to examine molecular interactions of the amyloid precursor protein APP in Drosophila models of Alzheimer´s disease.

Author response: The above review is very interesting and important to increase our understanding of APP molecular function and interactions but will not be included since our review is focused on proteotoxicity of the Aβ peptide.

Recently, also many new findings have been published on the molecular mechanisms and mediators of neuroprotection (see for instance but not only):

Tsakiri et al., 2021 Amyloid toxicity in a Drosophila Alzheimer´s model is ameliorated by autophagy activation.

Author response: This article examines the role of proteostatic or antioxidant pathways in AD using an AβPP-BACE1 Drosophila model of AD. It will not be included in our review since we are focusing on Drosophila models of AD where various Aβ peptides are expressed directly from the transgene.

Also classical work like that of Chakraborty et al., 2011, Characterization of a Drosophila alzheimer´s disease model: pharmacological rescue of cognitive defects. needs to be discussed.

Author response: This reference has been added in the review in line 259.

Thank you for refering to these papers and the many important references cited therein. Please discuss the hot topics therein briefly in addition to the other aspects already covered.

Specific suggestions: 

Line 279: Error! Reference source not found. This bold insert can be deleted.

Author response: We cannot see this error in the manuscript.

Line 442: Same as above

Author response: We cannot see this error in the manuscript.

Reviewer 2 Report

ijms1377379: Exploring Abeta Proteotoxicity and Drug Candidates Using Drosophila 
melanogaster  

This review is one of the best recent ones about the Alzheimer's disease (AD) research field, using Drosophila melanogaster as a model system. The figures are excellent and useful. Those who are intending to start AD research should read this review to explore the research strategy, selecting the research system among culture cells, mice, C. elegans, or D. melanogaster. This review is also useful to the experts of the AD field to cover the current D. melanogaster research for AD. The manuscript is almost complete and can be published in the present form. As some points, which may improve the value of this review a little bit, are suggested as follows, the decision is "accept after minor revision". However, this manuscript is almost "accept in present form."

<Minor points>
(1) D. melanogaster is not a human patient.
It may be difficult to discuss. However, the comprehensive understanding about the difference between human patients and experimental models is always a very important point, because only little results observed in experimental models can be directly applied to human patients. For example, those substances which are effective in experimental models are seldom valid in clinical trials. In that sense, the limitation or precaution to extend the research results of D. melanogaster to the human disease should be briefly discussed. For example, the expressed human protein is surrounded by D. melanogaster proteins. This is a very unusual condition.

(2) Dementia of D. melanogaster
Although the natural dementia of D. melanogaster is not a main matter of this review, the dementia without Abeta is another important point of view to elucidate human dementia. Especially, the pathologies of elderly AD are not only those related to Abeta. There may be vascular and metabolic malfunctions in the elderly brain. Whether or not there is a possibility that overexpressed Abeta not only had direct toxicity to neurons but also accelerated natural dementia in D. melanogaster. In addition, the elderly D. melanogaster will never have senile plaques in the brain. Brief discussion about this matter may help the understanding of dementia beyond Abeta.

(3) A table for the comparison of APP (APP related molecules of D. melanogaster), secretases, and so on between human and D. melanogaster
There are some differences in the proteins directly related to APP and its processing between humans and D. melanogaster. A summary table is useful.

(4) Some compiling or command error may have happened.
line 279: is illustrated in Figure 2<Error! Reference source> line 280: <not found.>. Unravelling the importance of different
line 442: <Error! Reference source not found.>Figure 3 shows an overview of the

End of File

Author Response

Reviewer 2:

This review is one of the best recent ones about the Alzheimer's disease (AD) research field, using Drosophila melanogaster as a model system. The figures are excellent and useful. Those who are intending to start AD research should read this review to explore the research strategy, selecting the research system among culture cells, mice, C. elegans, or D. melanogaster. This review is also useful to the experts of the AD field to cover the current D. melanogaster research for AD. The manuscript is almost complete and can be published in the present form. As some points, which may improve the value of this review a little bit, are suggested as follows, the decision is "accept after minor revision". However, this manuscript is almost "accept in present form."

Minor points

(1) D. melanogaster is not a human patient.
It may be difficult to discuss. However, the comprehensive understanding about the difference between human patients and experimental models is always a very important point, because only little results observed in experimental models can be directly applied to human patients. For example, those substances which are effective in experimental models are seldom valid in clinical trials. In that sense, the limitation or precaution to extend the research results of D. melanogaster to the human disease should be briefly discussed. For example, the expressed human protein is surrounded by D. melanogaster proteins. This is a very unusual condition.

Author response: To address the issue stated above the text in the manuscript has been adjusted. Line 464: “Although, there are differences in the proteome between a fly brain and a human brain that might limit the possibility to directly apply results from fly experiments to humans, the fly has proven to be a powerful tool to unravel Aβ related proteotoxicity and to find potential drug candidates.”

(2) Dementia of D. melanogaster
Although the natural dementia of D. melanogaster is not a main matter of this review, the dementia without Abeta is another important point of view to elucidate human dementia. Especially, the pathologies of elderly AD are not only those related to Abeta. There may be vascular and metabolic malfunctions in the elderly brain. Whether or not there is a possibility that overexpressed Abeta not only had direct toxicity to neurons but also accelerated natural dementia in D. melanogaster. In addition, the elderly D. melanogaster will never have senile plaques in the brain. Brief discussion about this matter may help the understanding of dementia beyond Abeta.

Author response: The above stated issue is a very interesting topic but is beyond the scope of our review and will not be included.

(3) A table for the comparison of APP (APP related molecules of D. melanogaster), secretases, and so on between human and D. melanogaster. There are some differences in the proteins directly related to APP and its processing between humans and D. melanogaster. A summary table is useful.

Author response: Since our review in focused on proteotoxicity of various Aβ peptides the above suggested summary table will not be included.

(4) Some compiling or command error may have happened.
line 279: is illustrated in Figure 2<Error! Reference source> line 280: <not found.>. Unravelling the importance of different
line 442: <Error! Reference source not found.>Figure 3 shows an overview of the

Author response: We cannot see these errors in the manuscript.

Reviewer 3 Report

The manuscript “Exploring Aβ Proteotoxicity and Drug Candidates Using Drosophila melanogaster” by Elovsson et al. is a review on studies performed in Drosophila melanogaster addressing potential target mechanisms for Alzheimer’s disease therapy.

The review is exhaustive, comprehensible, and complete.

Limitations of the approaches based on the Drosophila model should be mentioned in the conclusion section.

As a suggestion, I’d modify the term “drug candidates” in the title since this is a specific term in drug discovery which does not suitably apply to this model.

At lines 279 and 442 an error message is displayed.

Author Response

Reviewer 3:

The manuscript “Exploring Aβ Proteotoxicity and Drug Candidates Using Drosophila melanogaster” by Elovsson et al. is a review on studies performed in Drosophila melanogaster addressing potential target mechanisms for Alzheimer’s disease therapy.

The review is exhaustive, comprehensible, and complete.

Limitations of the approaches based on the Drosophila model should be mentioned in the conclusion section.

Author response: The text has been adjusted accordingly. Line 464: “Although, there are differences in the proteome between a fly brain and a human brain that might limit the possibility to directly apply results from fly experiments to humans, the fly has proven to be a powerful tool to unravel Aβ related proteotoxicity and to find potential drug candidates.”

As a suggestion, I’d modify the term “drug candidates” in the title since this is a specific term in drug discovery which does not suitably apply to this model.

Author response: The title has been adjusted accordingly “Exploring Aβ Proteotoxicity and Therapeutic Candidates Using Drosophila melanogaster”.

At lines 279 and 442 an error message is displayed.

Author response: We cannot see these errors in the manuscript